# Global patterns and drivers of fish reproductive potential on coral reefs

Jeneen Hadj-Hammou [1] ✉, Joshua E. Cinner [2], Diego R. Barneche [3,4], Iain R. Caldwell [5], David Mouillot [6], James P. W. Robinson [1], Nina M. D. Schiettekatte [7], Alexandre C. Siqueira [8,9], Brett M. Taylor [10] & Nicholas A. J. Graham [1]

Fish fecundity scales hyperallometrically with body mass, meaning larger females produce disproportionately more eggs than smaller ones. We explore this relationship beyond the species-level to estimate the "reproductive potential" of 1633 coral reef sites distributed globally. We find that, at the site-level, reproductive potential scales hyperallometrically with assemblage biomass, but with a smaller median exponent than at the species-level. Across all families, modelled reproductive potential is greater in fully protected sites versus fished sites. This difference is most pronounced for the important fisheries family, Serranidae. When comparing a scenario where 30% of sites are randomly fully protected to a current protection scenario, we estimate an increase in the reproductive potential of all families, and particularly for Serranidae. Such results point to the possible ecological benefits of the 30 × 30 global conservation target and showcase management options to promote the sustainability of population replenishment.

Coral reefs are some of the most biodiverse ecosystems in the world and provide essential goods and services to millions of people[1,2]. However, anthropogenic disturbances have caused these ecosystems to rapidly degrade[3]. Marine Protected Areas (MPAs) and fishing restrictions are some of the most widely applied management tools used to mitigate against such disturbances[4–6]. One major benefit conferred by protection from fishing is that it enhances fish biomass, which can increase certain ecosystem functions and services[7–9].

At the community-level, biomass production is a product of the dynamic processes of reproduction, recruitment, growth, and mortality[10]. Each of these processes are impacted by variables that act across scales. For example, fish communities on coral reefs are made up of hundreds of species with varying reproductive traits such as

fecundity, length at maturity, and length at sex change. These traits are differentially impacted by disturbances like fishing and climate change[11]. Some lab-based experiments have shown that fishing can lead to genetic shifts in fecundity[12] and temperature may have strong impacts on fish size and size-dependent reproductive traits[13,14]. Fishing can also cause larger-scale shifts in population sex-ratios[15,16] or the age and size structure of fish communities[17,18], which in turn affects which fish are able to reproduce and the number of eggs produced, respectively[19,20]. At the individual-level, large, old, female fish produce more, potentially higher quality eggs than smaller, younger females[21]. Egg production can also be corelated with recruitment success[22]. However, large, old, female fish often represent a small proportion of total fish community biomass, and it has been demonstrated that for

[1]Lancaster University Environment Centre, Lancaster University, Lancaster, UK. [2]Thriving Oceans Research Hub. School of Geosciences, University of Sydney, Caperdown, NSW 2006, Australia. [3]Australian Institute of Marine Science, Crawley, WA, Australia. [4]Oceans Institute, The University of Western Australia, Crawley, WA, Australia. [5]College of Arts, Society and Education, James Cook University, Townsville, QLD, Australia. [6]MARBEC, Univ Montpellier, CNRS, IRD, Ifremer, Montpellier, France. [7]Hawai'i Institute of Marine Biology, University of Hawai'i at Mānoa, Kāne'ohe, HI, USA. [8]Centre for Marine Ecosystems Research, School of Science, Edith Cowan University, Perth, WA 6027, Australia. [9]Research Hub for Coral Reef Ecosystem Functions, College of Science and Engineering, James Cook University, Townsville, QLD 4811, Australia. [10]University of Guam Marine Laboratory and UOG Sea Grant, 303 University Drive, UOG Station, Mangilao, Guam 96923, USA. ✉e-mail: jeneenhh@gmail.com

some species, numerous, young, mature, female fish make up more significant contributions to larval replenishment[23].

Within a fisheries management paradigm, "Total Egg Production" is generally regarded as a better measure of reproductive potential than "Spawning Stock Biomass"[24]. While there are many factors influencing the pathway from community reproductive potential to biomass production (e.g. larval survival[25], habitat availability[26], etc.), protection from fishing is likely to enhance both reproductive potential and biomass production at independent points of the pathway[27]. Protected areas can enhance fish larval supply while connectivity between reserves helps to ensure long-term population sustainability by enhancing recruitment and population replenishment[22,28,29]. Reproductive potential is also essential to maintaining "compensatory buffering productivity" within heavily fished sites with low biomass levels[30], although total landings may be reduced due to lost fishing grounds after MPA establishment[31] or if larval dispersal is limited[32].

The reproductive contribution of fish inside protected areas has historically been underestimated[27]. It had previously been assumed that fecundity scaled isometrically with female fish mass. However, Barneche et al. in ref. 33 demonstrated that, on average, fecundity scales hyperallometrically (with an average exponent of 1.18) with fish size, meaning that larger fish produce disproportionately more eggs than smaller fish.

Although this relationship between fecundity and body mass has been established at the species level, the global patterns of fish reproductive potential on coral reefs and the drivers of these patterns are still unknown. To address this critical gap, we use the hyperallometric fecundity-mass scaling model and its uncertainty to estimate the reproductive potential of fish across 1633 coral reef sites distributed globally. We define "reproductive potential" in this paper as an estimate of the potential combined species' total egg production at a snapshot in time. To obtain this measure, we first estimate the biomass of mature females at each site by extrapolating phylogenetic Bayesian regression models on species lengths at maturity and sex ratios. We then estimate reproductive potential as a snapshot of the batch fecundity of all mature female fish on a reef by extrapolating the fecundity model developed in ref. 33 to 831 reef fish species, based on high resolution phylogenetic trees from Siqueira et al. in ref. 34. In doing so, we elaborate on the use of Spawning Stock Biomass and Total Egg Production as a fisheries' proxy for reproductive potential[35] to incorporate more complexity through population-level fecundity outputs accounting for variation in sex-ratios, length at maturity, and associated modelled uncertainties. We investigate how this potential varies across socio-ecological gradients and simulate the impact of implementing protected areas in line with global 30% protection targets[36,37]. This approach allows us to ask questions on a coarse level to enable the detection of large-scale trends, and to understand how the relationship between fecundity and biomass scales beyond the species level. Outlining these trends can contribute to conceptualising the process of biomass production and further our understanding of the conditions that facilitate another important fisheries and conservation goal, sustained population replenishment.

## Results
### Global reproductive potential estimates
Our results show the reproductive potential (or estimated batch population fecundity) of 1633 reef sites distributed across 35 countries, states, or territories, and 4 marine realms. The reproductive potential of all fish at the reef site ranged from 12,337,945 to 443,376,565 eggs/ha (Fig. 1A; Fig. 1B; Supplementary Fig. 1). At the marine realm level, fully protected areas in the Western Indo-Pacific had the highest median population fecundity (19.28 log eggs/ha, 95% Uncertainty Interval (UI):19.07–19.41; $n = 40$ sites), followed by fully protected areas in the Tropical Atlantic (19.25 log eggs/ha, 95%UI: 18.96–19.32; $n = 21$ sites). Fished sites in the Western Indo-Pacific had the lowest median population fecundity (17.29 log eggs/ha, 95%UI:

16.82–18.96; $n = 57$ sites), followed by fished sites in the Eastern Indo-Pacific (17.61 log eggs/ha, 95%UI: 16.67–18.52; $n = 500$ sites) (Fig. 1A; Supplementary Table 1). At the nation/state/territory level, restricted sites in Fiji had the highest median fecundity (19.61 log eggs/ha, 95%UI: 16.61–22.59; $n = 1$ site), while fished sites in Guam had the lowest median fecundity (16.60 log eggs/ha, 95% UI: log 13.70–19.50; $n = 8$ sites) (Fig. 1B). Log fecundity calculated at the reef site level scaled with log mature female biomass with a slope of 1.07 (95%UI: 1.02–1.12; $R^2 = 0.98$, 95%UI:0.98–0.99) (Fig. 1C).

### Global drivers of reproductive potential
We assessed if previously hypothesised global drivers of reef fish biomass and ecological functions[38,39] would also explain mature female biomass, the reproductive potential of all surveyed fish families, and the reproductive potential of three economically important families: Lutjanidae, Labridae (Scarini), and Serranidae. We constructed Bayesian hierarchical models to determine the effect of 11 socio-ecological indicators: gravity (an estimate of human pressure based on population size and travel time to the fishing site[14]; see methods), local human population growth, reef fish landings, human population size, protection, ocean productivity, climate stress, reef habitat, depth, sampling method, and sampling area.

Examining the impact of protection, we find that fully protected areas had the greatest positive effect on the fecundity of all fish families (0.72, 95%UI:0.32–1.13), followed by fishing restrictions (0.38, 95%UI:0.13–0.63) (Fig. 2). Fully protected areas and fishing restrictions had a positive effect on Labridae (Scarini) (0.76, 95%UI:0.36–1.17; 0.43, 95%UI:0.18-0.69), and Serranidae (1.71, 95%UI:0.93–2.48; 0.84, 95% UI:0.37–1.30) fecundity, but neither form of protection had an effect on Lutjanidae (0.47, 95%UI:−0.12–1.08; −0.02, 95%UI:−0.42 to −0.40). Total gravity had the greatest negative effect on fecundity of all fish families (−1.02, 95%UI:−1.24 to −0.80), as well as a strong negative impact on Lutjanidae (−0.48, 95%UI:−0.90 to −0.07), Labridae (Scarini) (−0.62, 95%UI:−0.90 to −0.33), and Serranidae (−0.92, 95%UI:−1.37 to −0.45) (Fig. 2). Reef fish landings had the greatest negative effect on Serranidae fecundity (−2.21, 95%UI:−3.90 to −0.51). Methodology and habitat type also strongly influenced results. Distance sampling had the greatest positive effect on the fecundity of all families (1.57, 95% UI:0.95–2.17), while depth (< 4 m) had the greatest negative effect on Lutjanidae fecundity (−0.64, 95%UI:−0.98 to −0.30), but we control for sampling method, habitat type, and depth in the comparison of fecundity values across geographies and protections by setting all these variables to their reference levels (e.g. Fig. 1A and Fig.1B).

We replicated the model of fish biomass >20 cm from Cinner et al. in ref. 39, using the sites selected for this study, for comparison to fecundity. The ratios of fully protected/fished and restricted/fished posterior draws from global drivers models of fish with a biomass >20 cm, mature female fish biomass, fecundity of all fish families, fecundity of Lutjanidae, fecundity of Labridae (Scarini), and fecundity of Serranidae are depicted in Fig. 3. Biomass of fish >20 cm in fully protected areas was 2.37 (95%UI:1.35–4.14) times higher than in fished areas. Mature female biomass in fully protected areas was 1.74 (95% UI:1.15–2.64) times higher than in fished areas. The fecundity of fish in fully protected areas was 2.05 (95%UI:1.27–3.31) times higher than in fished areas. However, greater differences between the fecundity of fished and fully protected sites were observed for key target fisheries families. The highest ratio between fecundity of fully protected/fished and restricted/fished sites was for the family Serranidae; fecundity was 5.46 (95%UI:2.53–11.9) times greater in fully protected sites compared to fished sites and 2.31 (95%UI:1.45–3.68) times higher in restricted sites than unrestricted sites (Fig. 3).

### Reproductive potential gains from 30% protection
The UN Biodiversity Conference of the Parties 2022 established a global target to achieve "Effective conservation and management of at

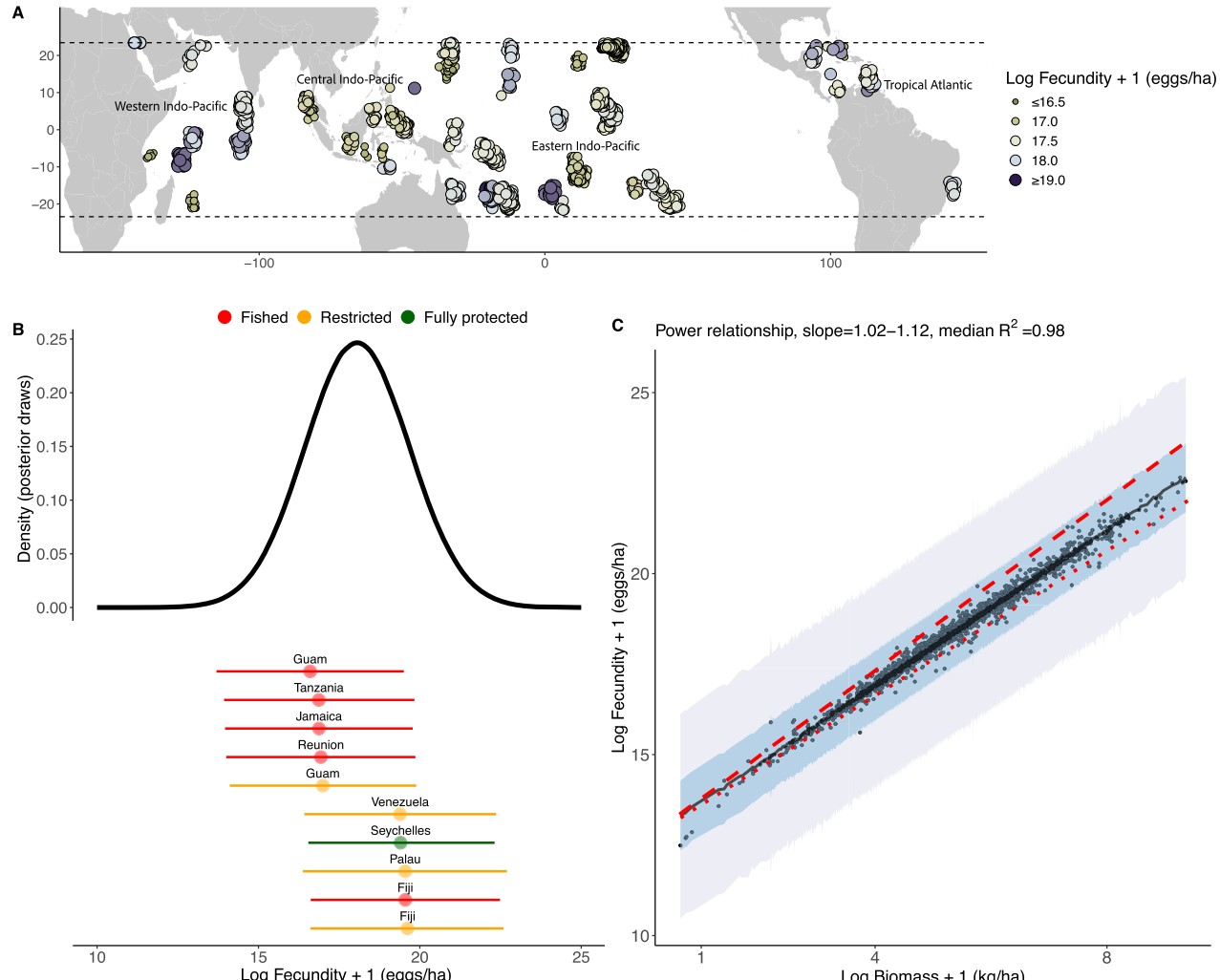

**Fig. 1 | Global patterns in coral reef fish fecundity. A** Global map of fish fecundity (log +1 eggs/ha) on coral reefs across all sites (*n* = 1633 sites; standardised across methodology, habitat, and depth), ranging from ≤ 16.5 (small circle, dark brown) to ≥ 19.0 (log +1 egg/ha) (large circle, dark blue). Marine realms from ref. 94 are identified on the map in text. **B** Distribution of coral reef fish fecundity (log +1 eggs/ha) values across all sites (*n* = 4000 posterior draws for each of the 1633 sites, standardised across methodology, habitat, and depth). Points highlighted below the density curve represent the five highest (1. Fiji Restricted *n* = 1 site; 2. Fiji Fished *n* = 15 sites; 3. Palau Restricted *n* = 2 sites; Seychelles Fully Protected *n* = 3 sites; 5. Venezuela Restricted *n* = 7 sites) and five lowest (1. Guam Restricted *n* = 4 sites; 2.

Reunion Fished *n* = 14 sites; 3. Jamaica Fished *n* = 8 sites; 4. Tanzania Fished *n* = 6 sites; 5. Guam Fished *n* = 8 sites) median fecundity values of the distribution, calculated by country/protection regime, with 95% Uncertainty Intervals (UI) depicted as horizontal bars. Points are colored by protection level, with fished sites in red, restricted fishing sites in orange, and fully protected area sites in green. **C** Linear model of log fecundity -log biomass, illustrating a power law relationship, with a slope of 1.03–1.13 and R² = 0.98 (*n* = 4000 posterior draws). Points are raw values (*n* = 1633 sites). Black line indicates median modelled relationship, with 95% (purple) and 50% (blue) UI shaded. Slopes of 1 (isometric relationship) and 1.18 (as estimated in ref. 33) are indicated as red dotted and dashed lines, respectively.

least 30 per cent of the world's land, coastal areas and oceans" by 2030[36]. Using our Bayesian hierarchical model of global drivers of reproductive potential, we simulated the potential fecundity gains associated with establishing 30% protection (hypothetical fully protected areas) across four marine realms compared to current protection levels. We calculated the current percentage of protection across marine realms by estimating the percentage of coral reef area from the Allen Coral Atlas[40] that overlapped with fully protected (no-take) MPAs from the World Database of Protected Areas[41]. Simulated datasets therefore consisted of the 1633 sites with protection status changed to reflect the scenario and all other covariates held to their means or reference levels.

When examining all fish families together, the Tropical Atlantic was predicted to have the highest median increase in fecundity, gaining 32.52% (95%UI: 19.92–47.15) - from 18.36 (log eggs/ha; 95%UI: 17.19–19.16) in the current levels of protection scenario (0.77% protected), to 18.68 (log eggs/ha; 95%UI: 17.53–19.43) in the 30%

protection scenario. The Central Indo-Pacific (currently 5.76% protection) was predicted to have the lowest median increase in fecundity, gaining 24.34% (95%UI: 13.84–35.44) - from 18.15 (log eggs/ha; 95%UI: 16.88–19.05), to 18.37 (log eggs/ha; 95%UI: 17.10–19.27). We also explored the percent gains in Serranidae fecundity, as full protection had the strongest effect on this family (Figs. 2, 3). We found that for all marine realms except for the Central Indo-Pacific, median predicted percent gains were above 100%, suggesting a minimum of double gain in most regions. Again, the Tropical Atlantic was predicted to have the highest percent gain in fecundity (129.35; 95%UI: 82.75–184.73), and the Central Indo-Pacific was predicted to have the lowest percent gain in fecundity (88.20%; 95%UI: 51.98–131.64) (Fig. 4).

## Discussion

Fully protected areas and fishing restrictions had a positive effect on the modelled reproductive potential of coral reef fishes. In particular, fully protected areas conferred a great advantage to the reproductive

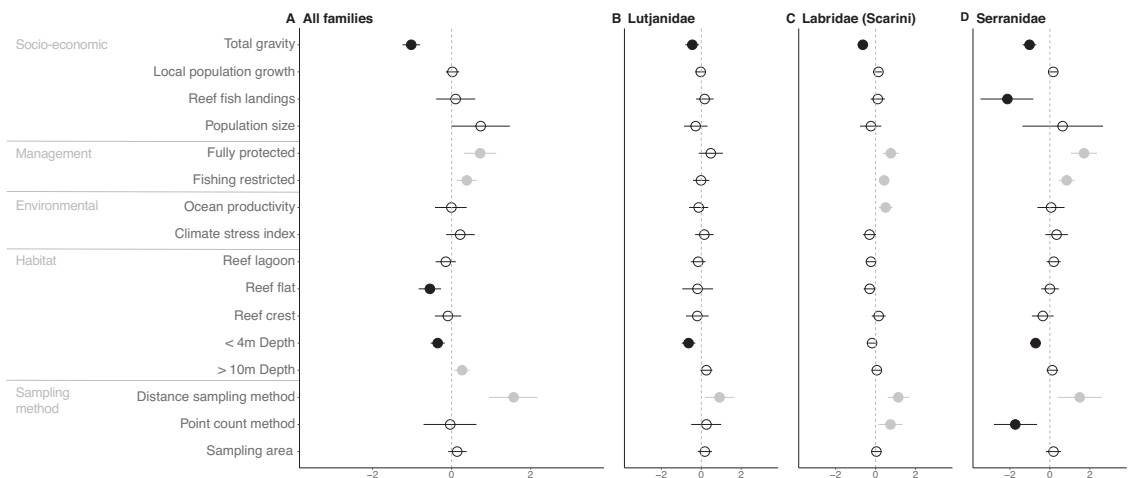

**Fig. 2 | Drivers of coral reef fish fecundity.** Median standardised effect size and 95% UI of predictors (*n* = 4000 posterior draws) on **A** fecundity of all families, as well as the fecundity of economically important families including **B** Lutjanidae, **C** Labridae (Scarini), and **D** Serranidae. Points are coloured black for a negative effect size, grey for a positive effect size, and blank for an effect size that overlaps with 0.

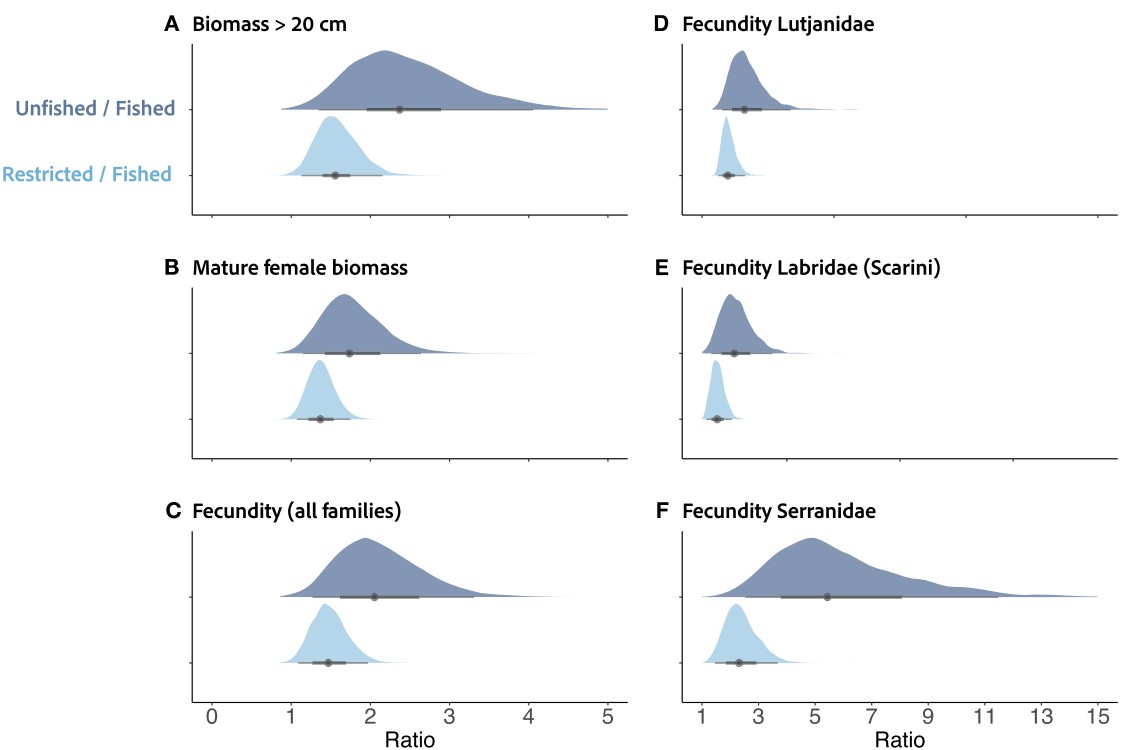

**Fig. 3 | Effects of protection on coral reef fish biomass and fecundity.** Ratios of draws from the posterior distributions (*n* = 4000 posterior draws) of fully protected sites/fished sites (dark blue) and restricted sites/fished sites (light blue) for models of **A** fish with a biomass > 20 cm, **B** mature female fish biomass, **C** fecundity of all fish families, **D** fecundity of Lutjanidae, **E** fecundity of Labridae (Scarini), and **F** fecundity of Serranidae. Medians are illustrated as points with 95% (thick lines) and 50% (thin lines) UI.

potential of economically important fisheries species, such as those in the family Serranidae. The modelled reproductive potential of this family inside fully protected areas was 5.46 (95%UI: 2.53–11.90) times higher than in fished areas. Reef fisheries landings also had a strong negative effect on Serranidae fecundity, highlighting its value as a target fisheries family, including in restricted fishing areas. Our model simulations show that if 30% protection were achieved, there could be an increase of up to 129% (95%UI: 82.75–184.73%) in the reproductive potential of Serranidae across marine realms, and an increase of up to 33% (95%UI: 19.92–47.15) in the reproductive potential of all fish families when pooled together. With all these results, it is critical to interpret median estimates within the context of their uncertainty and with the knowledge of our model assumptions and limitations. This is particularly important for conservation practitioners and policy makers wishing to assess the potential impact of protection, as our models make a series of assumptions (discussed below) and our methods were not able to account for a range of variables which are known to influence the success of protection (e.g. the size and age of

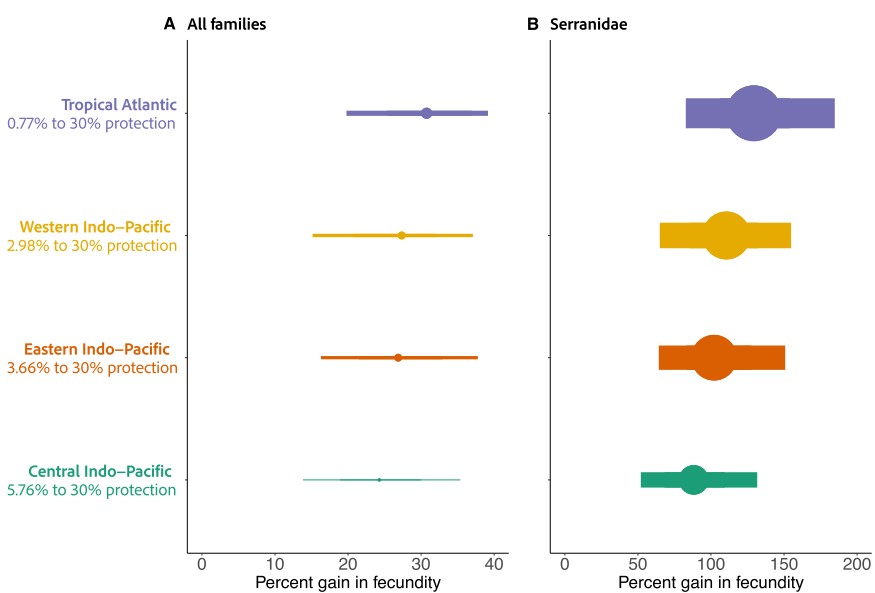

**Fig. 4 | Potential coral reef fish fecundity gains with 30% protection.** Potential percent fecundity gains (medians and 95%UI) to be made from increasing currently estimated protection levels across marine realms [Tropical Atlantic ($n$ = 99 sites) = 0.77%, Western Indo-Pacific ($n$ = 223 sites) = 2.98%, Eastern Indo-Pacific ($n$ = 622 sites) = 3.66%, Central Indo-Pacific ($n$ = 689 sites) = 5.76%] to 30% protection in **A** all fish families, and **B** Serranidae. Point size and line thickness are scaled to percent gain in fecundity to highlight scale differences between (**A**) and (**B**).

protected areas, compliance with regulations, types of fishing restrictions, etc.[4,42]).

Contrary to patterns observed at the level of individual species[27], the ratio of modelled reproductive potential of all fish in fully protected/fished sites and restricted/fished sites was lower than or similar to the ratio of modelled biomass (biomass of fish >20 cm and mature female biomass respectively) in fully protected/fished sites and restricted/fished sites. This was an unexpected finding because, at the species level, fecundity was found to scale with female biomass with a median exponent of 1.19 (95% UI: 1.11–1.26; see methods section on modelling "species fecundity"). In light of this relationship, we expected previously observed large differences in biomass between fished and fully protected sites[38] to be exponentially larger for population batch fecundity. However, we demonstrate the compounding importance of community composition in driving the difference in patterns between biomass of fish >20 cm, mature female biomass, and population fecundity. As per our methods, the calculation for biomass of fish >20 cm is primarily dependent on community size structure and filters out all fish below 20 cm, including those that might be mature females. The calculation for mature female biomass depends on community size structure, sex ratios, and lengths at maturity. Reproductive potential calculations account for differences in community size structure, sex ratios, lengths at maturity, and species-level fecundity. Protection had a positive effect on biomass, but there was no difference between management regimes when examining length at maturity and species-level fecundity (See Supplementary Fig. 2; Supplementary Fig. 3). Thus, a protected site could have a high biomass but be comprised of poorly fecund individuals or many immature individuals, while a fished site could display the opposite pattern, and vice versa. When the variation is reduced, for example, by focussing on individual families such as Serranidae, we can highlight greater differences between the reproductive potential of protected and fished areas.

We also found that the strong hyperallometric relationship between fecundity and biomass observed at the species-level is dampened when comparing sites at a larger spatial scale. At the global level, across all species, there was a log-log linear relationship between fecundity and biomass with a slope of 1.07 (95%UI: 1.02–1.12), compared to the species-level slope of 1.19. Once again, this could be

attributed to the community composition of sites. Looking at the averaged effect across divergent communities dampens the relationship between fecundity and mature female biomass. This highlights that reproductive potential is not only linked to biomass, but also depends on community composition and population size structure, both of which vary substantially at the global scale[43].

This dampened effect of protection across families is reflected in differences between the reproductive potential gains under the 30% protection scenario for all families compared with Serranidae, a key fisheries target family. Across marine realms, a maximum of a 33% (95% UI: 19.92–47.15) increase in reproductive potential across families was possible. Comparatively, we found that there could be up to a 129% (95%UI: 82.75–184.73%) gain in fecundity for Serranidae. However, these scenarios were produced by randomly sampling sites to be protected and assuming fishing pressure was not displaced through leakage towards fishing grounds surrounding MPAs[44]. The location and selection of protected areas is critical to their success, both ecologically and socially[45–47]. For example, Fontoura et al. in ref. 29 demonstrate the importance of the location of MPAs to ensure larval "connectivity conservation". They suggest that there could be a disproportionately positive effect on the persistence of ecosystem service provision by implementing protections in biodiversity hotspots with key dispersal corridors, larval sources, and sinks. Future work can integrate such larval connectivity models with data on reproductive potential to further maximise conservation and ecosystem service gains, targeting areas based on knowledge of community composition and local socio-environmental contexts.

It is possible that given the magnitude of difference between the modelled reproductive potential of fish inside and outside protected areas, larval export benefits to fisheries could be substantial[48–50]. However, our models do not account for population dynamics[51] and reproductive compensation mechanisms[52]. The ways in which fishing affects the abundance and size structure of coral reef fish through density-dependence mechanisms varies greatly across species, making it challenging to analytically incorporate the effects of density-dependence in highly diverse coral reef ecosystems[53,54]. In some density-dependent systems, the increasing density of fish can result in a reduction of the number of eggs produced per individual[52]. Given this

mechanism, it is possible that individuals recorded within our surveys, especially in high abundance sites within protected areas, would have a lower reproductive output than suggested by our models. Moreover, if fishing induced evolution caused species to compensate for size truncation by maturing earlier[55], our models could have underestimated the reproductive potential of some fished sites. However, these effects are poorly quantified and highly variable between individuals, species, and habitats, which could mean that they become negligible when species are pooled together in a large-scale analysis. These limitations to our approach mean that our results should be interpreted as an indication of the possible effects of conservation actions, rather than guaranteed outcomes.

More comprehensive evidence quantifying the interacting effects of fishing and density-dependence on the reproductive traits of a diversity of species is needed to better improve estimations of reproductive potential. Future work could also account for differences in species' life history/reproductive strategies that influence survival (e.g. reproductive care strategies) and lifetime reproductive output[56–58]. Accounting for such variables will further increase the variation observed in reproductive potential between sites, but it could also increase our estimates of the impact of protection, particularly if protection has a direct impact, rather than an impact mediated through just biomass.

Protected areas are a critical management tool that can produce multiple related and co-occurring socio-ecological benefits[59]. However, coral reefs are complex ecosystems, and our work highlights that the magnitude of impact from protection can vary greatly between conservation or fisheries management goals. Fully protected areas can be challenging to implement and can lead to creating local inequities through restricting access to important resources needed for local livelihoods[60]. Fishing restrictions, on the other hand, can be easier and more equitable to establish, and have been shown to produce positive fisheries and conservation outcomes[61–63]. Ultimately, trade-offs between the benefits and losses of management actions can only be considered with a full understanding of what is feasible and desirable in a local context. While our work takes a large-scale simulation approach to understand the potential value of increasing global protection coverage to 30% on coral reefs, Sandbrook et al. in ref. 64 emphasise that, ultimately, such targets are going to be implemented at the national and sub-national scales and therefore more work is needed to understand the effects of protection at a policy implementation-relevant scale. We recommend extending the approaches developed here to calculate the reproductive potential at such scales. Furthermore, future work will need to bridge the gaps between the subsequent stages of biomass production to move beyond the static snapshot statistic of potential batch fecundity, to a dynamic understanding of reproduction and recruitment.

## Methods
### Reef fish survey methods
Research permits for ecological surveys were obtained from relevant country or territory authorities. Ethical approvals were not required as the research was purely observational. A total of 4089 surveys conducted between 2004 and 2013 from 1633 tropical reef sites across 35 nations, states, or territories were included in this study. Surveys used either belt transect, distance sampling, or point count methods to identify fishes to species level and estimate their total length and abundance. The belt transect and point count methods involve surveying fish within a fixed area, either along a transect line with a fixed width (belt transect), or within a circular area from a fixed point (point count). The distance sampling method however does not have a fixed survey area. Instead, the observer records the perpendicular distance of the fish from either a transect line or a stationary point, and these distances are used to calculate the total surveyed area[65–67].

We excluded cryptobenthic reef fish, sharks, and semi-pelagic species. For each reef site, habitat type (slope, crest, flat, lagoon/back reef), depth range (0–4 m, 4–10 m and >10 m), and total sampling area were recorded. Details about these surveys can be found in ref. 38.

### Scales of data
We separated the data into four nested scales (listed from smallest scale to largest scale):
1. **Surveys** = as specified above.
2. **Reef site** = aggregations of replicate surveys within a few hundred meters (mean of 2.4 surveys/site).
3. **Reef cluster** = reef sites within 4 km of each other were clustered together as specified in the methodology of Cinner et al. in ref.[39]. Social and environmental covariates of the global drivers models were estimated at this scale.
4. **Nation/state/territory.**

### Statistical analyses
Fecundity is defined as the number of eggs produced per mature female in a single spawning event[33]. In this paper, we scale fecundity up to the level of the population, and define population fecundity, or reproductive potential, as the total number of eggs produced by all mature female fish per hectare[68]. Underwater Visual Count survey data (including belt transect, distance sampling, and point count methods) records the number, taxonomic identity, and total length of fish. We applied a five-step process to get from this form of data to an estimate of population fecundity for each site: estimate 1) biomass, 2) fecundity, 3) sex ratios, and 4) length at maturity of each fish, and then 5) calculate the mature female biomass and population fecundity at each site. Steps 2–4 all involved fitting a phylogenetic Bayesian regression in the R package "brms"[69]. The phylogenetic tree was obtained from Siqueira et al. in ref. 34. All global drivers models were run with 4 chains, each with 10,000 iterations and a warm-up of 9000 iterations. Phylogenetic models were run with 4 chains, 15,000 iterations and a warmup of 7500 iterations. All analyses and figures were produced in R 4.3.3[70].

### Phylogenetic extrapolation
We used the R package "ape"[71] to calculate a variance-covariance matrix of phylogeny to account for species non-independence in the models. We used the R package "picante"[72] to predict the phylogenetic effects for species without data and estimated response variables by combining the predicted phylogenetic effects with the intercepts and slopes of each model, as in refs. 73,74.

**Step 1. Biomass estimation.** The biomass of each fish was calculated using length-weight conversions from FishBase according to the equation: $W = a \times L^b$, where L is the median value of the 5 cm size bin for the total length of each fish recorded in the field, and a and b are species-specific length-weight coefficients[75].

**Step 2. Species fecundity.** We used the "picante" method described above to predict the phylogenetic effects for missing species in our dataset across 1000 draws from the posterior distribution of the fecundity model in Barneche et al. in ref. 33. These effects were then combined with global intercepts and slopes to generate a distribution of 1000 possible fecundity values for each species/biomass combination, thereby accounting for the uncertainty in fecundity parameters. The fecundity model from Barneche et al. in ref. 33 is defined as:

$$\ln\text{Fecundity} = (\ln\alpha_0 + \ln\gamma_{0spp} + \ln\gamma_{0phy}) + (\beta_1 + \gamma_{1spp})*\ln\text{Biomass} + \ln\varepsilon \quad (1)$$

where lnFecundity is the natural log-transformed vector of fecundity values, $\ln\alpha_0$ is a fixed-effect intercept, $\ln\gamma_{0spp}$ and $\ln\gamma_{0phy}$ are respectively vectors of random-effect coefficients that account for residual intercept deviations attributable to species uniqueness and patterns of relatedness as described by the phylogeny, $\beta_1$ is a fixed-

effect slope for the natural log-transformed predictor vector, mature female mass, $\gamma_{1spp}$ is a vector of random-effect coefficients that account for residual slope deviations attributable to species uniqueness, and $\ln\varepsilon$ is the model unexplained residual variation. Fixed effects were assigned weakly informative priors following a Gaussian distribution, and random effects were assigned weakly informative priors following a Gamma distribution. We found that ln(fecundity) scaled positively with ln(biomass) with a slope of 1.19 (95% UI: 1.11–1.26). The phylogenetic heritability, estimated as the proportion of the variance (conditioned on the fixed effects) explained by the random effects (phylogeny) was 78.74% (95% UI: 64.14–89.50%).

**Step 3. Sex ratios.** Protogynous and protandrous species typically have sex ratios that deviate from 1:1[76], whereas gonochoristic species typically have close to 1:1 female to male sex ratios[77]. We therefore applied a 1:1 sex-ratio for gonochoristic species and collected data to model the sex ratios of non-gonochoristic species at the family level. The sexual pattern of each family was characterised based on[78,79].

We conducted a literature search using Google Scholar to obtain sex ratio data on five species with the highest proportional biomass of each protogynous or protandrous family. Where this data was not available, we expanded the search to ten species with the highest biomass, or where this information was not available, any other species from our species-list within that family, so that at least five sex ratio data points for each family were obtained (Supplementary Table 2). Sex ratio was then modelled as:

$$\text{Sex ratio} \sim \text{Beta}(N, \overline{p}, \text{phi})$$
$$\text{logit}(\overline{p}) = \alpha + \gamma \text{spp} + \gamma \text{phy} + \varepsilon \quad (2)$$

where $\alpha$ is a fixed-effect intercept, $\gamma_{spp}$ and $\gamma_{phy}$ are respectively vectors of random-effect coefficients that account for residual intercept deviations attributable to species uniqueness and patterns of relatedness as described by the phylogeny, and $\varepsilon$ is the model unexplained residual variation. Fixed effects were assigned weakly informative priors following a Gaussian distribution, random effects were assigned weakly informative priors following a Gamma distribution, and the phi precision parameter was assigned a weakly informative prior following the Gamma distribution. We calculated the Intraclass Correlation Coefficient of the model to assess phylogenetic heritability using variance decomposition methods[80]. The proportion of the variance explained by the grouping structure was 32.97% (95% UI: 24.36–42.00%).

We then sampled the dataset based on the sex ratio of each species by transect to select females. For example, if Species A was recorded 10 times in Transect 1, and the sex ratio of Species A was 6 females to 4 males, 60% of those records would be sampled randomly as females. We repeated this process 1000 times.

**Step 4. Length at maturity.** Temperature affects the life-history traits of fish[81]. As our dataset spans a large geographical scope, we wanted to account for the potential differences in the size at which species reach maturity across a temperature gradient. We therefore compiled raw and modelled data on the relationship between length at maturity (LMat) and sea surface temperature (SST). While it would have been better to include only raw data, rather than modelled data with its own errors and constraints, this data is not widely available. Our method captures the predicted relationship[82] and, crucially, accounts for an important source of variation in length at maturity across the dataset. Data were sourced from Morais and Bellwood in ref. 10, Morat et al. in ref. 83, Thorson et al. in ref. 84, and Wang et al. in ref. 81. We then modelled LMat as:

$$\ln\text{LMat} \sim \text{normal}(\mu, \sigma)$$
$$\mu = (\alpha + \gamma_{0spp} + \gamma_{0phy}) + (\beta + \gamma_{1spp}){}^{*}\text{SST} + \varepsilon \quad (3)$$

where $\alpha$ is a fixed-effect intercept, $\gamma_{0spp}$ and $\gamma_{0phy}$ are respectively vectors of random-effect coefficients that account for residual intercept deviations attributable to species uniqueness and patterns of relatedness as described by the phylogeny, $\beta$ is a fixed-effect slope for the predictor vector, SST, $\gamma_{1spp}$ is a vector of random-effect coefficients that account for residual slope deviations attributable to species uniqueness, and $\varepsilon$ is the model unexplained residual variation. Fixed intercepts and slopes were assigned weakly informative priors following a Gaussian distribution, and random effects were assigned weakly informative priors following a Gamma distribution. The phylogenetic heritability, estimated as the proportion of the variance (conditioned on the fixed effects) explained by the random effects (phylogeny) was 96.90% (95% UI: 94.45–98.10%).

We then used the NOAA Optimum Interpolation (OI) Sea Surface Temperature (SST) V2 product (https://psl.noaa.gov/data/gridded/data.noaa.oisst.v2.html) to calculate SST for all sites included in our dataset. The latitude and longitude coordinates corresponding to the centroid of each social site were assigned to the nearest coordinates available with SST data. The median date for fish survey data was 2008. We therefore used the mean annual SST values from 2003–2013 (2008 +/− 5 years). We extrapolated species' lengths at maturity using the "picante" method described above (see "Phylogenetic extrapolation" section) for the range of SSTs in which those species were recorded. We found an overall negative relationship between LMat and SST (slope = −0.05, 95% UI: −0.08 to −0.03).

**Step 5. Biomass of mature females and population fecundity.** The biomass of mature females and population fecundity at each site was estimated by first selecting all fish of the equal to or larger than the estimated length at maturity for that site. This was done across the 1000 samples of female fish (see "Sex ratios" section). The mean biomass of mature female fish and population fecundity at each site across the 1000 samples was calculated and converted into units of kg/ha and eggs/ha respectively to standardise across sampling area. We then stored the full distribution of site values for mature female biomass and fecundity and calculated the mean and standard deviation of the distribution of values for each site. This standard deviation was then used to quantify "measurement error" in further models, in order to propagate the uncertainty of the distribution of possible mature female biomass and fecundity values[85].

**Relationship between mature female biomass and population fecundity**

We used a Bayesian hierarchical mixed effects model to identify the parameters of the relationship between mature female biomass and population fecundity at the site level. We set reef cluster and nation/state/territory as random effects where reef sites are nested in reef clusters, and reef clusters are nested in nations/states/territories. In addition to capturing spatial variation, this hierarchical structure accounts for some of the variation across surveys arising from having different surveyors (inter-observer bias)[86], as each reef cluster is always represented by a single surveyor. Intra-observer bias in surveys is a source of variation which has some measurement error, but this is minimised through pre-survey training, calibration, and experience[87], all of which are represented by the observers of the data used in this study[38]. Population fecundity was therefore modelled as:

$$\ln\text{Fecundity}_{\text{Obs},0jk} \sim \text{Normal}(\ln\text{Fecundity}_{\text{True},0jk}, \ln\text{Fecundity}_{\text{StdError},0jk})$$
$$\ln\text{Fecundity}_{\text{True},0jk} \sim \text{Normal}\left(\mu_{0jk}, \sigma\right)$$
$$\mu_{0jk} = \alpha + \beta{}^{*}\ln(\text{mature female biomass}_{0jk}) + \gamma_{0jk} + \varepsilon \quad (4)$$

where $\alpha$ is a fixed intercept, $\beta$ is the slope, $\gamma_{0jk}$ is the matrix of random effect coefficients (reef cluster, nation/state/territory) that account for intercept variation, and $\varepsilon$ is the model unexplained residual variation. We used the random intercept values from the model to understand how each nation/state/territory deviated from the mean. We included a measurement error term for the response variable, where "Obs" corresponds to the fecundity values generated from preceding modelling steps with a standard error, "StdError". Including a measurement error around the response variable allowed us to propagate uncertainty through consecutive models without excessive computing power and time. This process causes "shrinkage", so that estimates for sites with high measurement error values were improved by pooling information from more certain estimates[85]. We used default "brms" model priors.

### Global drivers models

We elaborated on previous work investigating the global drivers of biomass[38,39] to assess if patterns were similar for mature female biomass, the fecundity of all families, and the fecundity of three economically important families: Lutjanidae, Labridae (Scarini), and Serranidae. We also replicated the model of fish biomass > 20 cm from Cinner et al.[39] for comparison to fecundity. The social and environmental drivers established in the previous works and incorporated into our models are:

1. **Management** = each *reef site* was assessed as being i) fully protected – a high compliance fully protected reserve, ii) restricted – active restrictions on gears or fishing efforts, or iii) openly fished – fished sites without any restrictions.
2. **Local human population growth** = the population growth of each *reef cluster* was calculated as the proportional difference between the population in 2000 and 2010, based on data from the Socioeconomic Data and Application Centre[88].
3. **Gravity** = an estimate of human pressure based on population size and travel time to the *reef site* from a population grid cell (see in ref. [89]).
4. **Human population size** = for each *nation/state/territory*, a population estimate for 2010 was derived from the national census reports CIA fact book (https://www.cia.gov/library/publications/the-world-factbook/rankorder/2119rank.html) and Wikipedia (https://en.wikipedia.org/wiki/Main_Page).
5. **National Reef Fish Landings** = data was obtained at the *nation/state/territory* level from the Sea Around Us Project (SAUP) catch database (http://www.seaaroundus.org). Estimates corresponding to 2010 and only including reef associated species were retained. Catch per unit area (catch/km²/y) was calculated by dividing a nation/state/territory's catch by its estimated reef area.
6. **Oceanic productivity** = average of monthly chlorophyll-a concentrations were calculated at the *reef cluster* scale using data provided at a 4km-resolution by Aqua MODIS (Moderate Resolution Imaging Spectro-radiometer) for years 2005 to 2010 as per Cinner et al. in ref. [39]
7. **Climate stress** = a measure of climate stress for corals developed by Maina et al. in ref. [90] was incorporated at the *reef site* scale.

We checked for collinearity between covariates using bivariate correlations and Variance Inflation Factors (VIF) and removed the Human Development Index (used in ref. [39]) from our models due to collinearity with reef fish landings (Pearson's r correlation > 0.6). All covariates included in the models had VIF scores less than 2. We then modelled biomass of fish >20 cm, mature female biomass, fecundity of all families, and the fecundity of specified families with the same hierarchical structure specified in Eq. [4], with reef cluster and nation/state/territory as random effects. All global drivers were scaled to a mean of zero (where continuous) and included in the model along with covariates accounting for methodological effects, sampling area,

census method, sampled habitat, and depth. In the models for biomass of fish > 20 cm, mature female biomass, and fecundity (all families and family-specific models), we took the natural log of response variables and the models were fit with a gaussian distribution error. For the family-specific models on fecundity, we present the results of models on non-zero values in the main text and the results of models incorporating zeros in the supplementary material (Supplementary Fig. 4). We modelled the full datasets (including zeros) using a hurdle-lognormal distribution to account for the large percentage of zeros. The two-part hurdle model was composed of 1) a binomial distribution to predict the probability of observing species in the specified family, and 2) a lognormal distribution of non-zero fecundity data. There is currently no way to incorporate a measurement error term on the response variable in models with a hurdle log-normal distribution. However, the results between the two sets of models are largely consistent (Fig. 2; Supplementary Fig. 4). We conducted graphical posterior predictive checks to assess all model fits to the data and ensured model convergence by checking trace plots and R-hat values (Supplementary Fig. 5–17)[69,91].

### Posterior estimates

Covariate effect sizes were visualised by sampling 4000 values from the posterior distributions of each model and estimating 50% and 95% uncertainty intervals. After identifying the importance of management type in all models, we estimated the ratio of biomass or fecundity between fully protected/fished and restricted/fished sites. We did this by holding all the covariates to their means or reference levels (thereby accounting for the large effect size of survey method and habitat), excluding the random effect structure, and allowing management to vary. We then sampled 4000 values from the posterior distributions of the model (using mean modelled predictions, i.e. the "posterior_epred()" brms function) from each management type and took the ratios of the posteriors.

### 30% protection scenario and current protection scenario

We calculated the current percentage of protection across marine realms by estimating the percentage of coral reef area in Allen Coral Atlas[40] that overlapped with fully protected (no-take) MPAs from the World Database of Protected Areas[41]. In order to compare the reproductive potential of a marine realm under current protection levels and under a 30% protection scenario, we created two sets of simulated data. Firstly, for the 30% protection scenario, we changed the management status of each site to fished, then we randomly sampled our sites 100 times and converted 30% of sites of each sample to protected. We then sampled 1000 draws from the posterior distribution of our model using the 100 simulated datasets and took the median value of each site-protection combination. We compared the median draws from this simulation to that of a current protection scenario. In the current protection scenario, rather than changing 30% of sites to protected, we sampled the dataset based on estimated levels of protection for each marine realm (i.e., Tropical Atlantic = 0.77%, Western Indo-Pacific = 2.98%, Eastern Indo-Pacific = 3.66%, Central Indo-Pacific = 5.76%). We illustrate the median percent difference in fecundity between current protection scenarios and 30% protection scenarios by marine realm.

### Reporting summary

Further information on research design is available in the Nature Portfolio Reporting Summary linked to this article.

## Data availability

All data required for data analyses are accessible via GitHub (https://github.com/Jeneen/ReproductivePotential) and the linked repository on Zenodo (https://doi.org/10.5281/zenodo.11528930)[92]. We accessed FishBase[75] to obtain length-weight conversions for fish biomass. We

used the covariates calculated in ref. 39 which include market gravity[38], local human population growth (https://sedac.ciesin.columbia.edu/), reef fish landings (http://www.seaaroundus.org), oceanic productivity (https://modis.gsfc.nasa.gov/data/), population size (https://www.cia.gov/library/publications/the-world-factbook/rankorder/2119rank.html; https://en.wikipedia.org/wiki/Main_Page), and climate stress[90]. The phylogenetic tree for phylogenetic extrapolation was obtained from ref. 34. Data for the model exploring temperature effects on length at maturity was obtained from[10,81,83,84]. Data used in the sex ratio model were obtained from sources outlined in Supplementary Table 2. We used the NOAA Optimum Interpolation (OI) Sea Surface Temperature (SST) V2 product (https://psl.noaa.gov/data/gridded/data.noaa.oisst.v2.html) to obtain SST values. We obtained sex ratio data by conducting a literature search on Google Scholar (citations for data sources are provided in Supplementary Table 2). The map for Fig. 1 used the "world" basemap provided by the R package "ggplot2"[93]. Coral reef area was calculated using Allen Coral Atlas[40] and the boundaries of fully protected (no-take) MPAs were obtained from the World Database of Protected Areas[41]. Source data are provided with this paper.

## Code availability

All code needed to reproduce analyses are available on GitHub (https://github.com/Jeneen/ReproductivePotential) and the linked Zenodo repository (https://doi.org/10.5281/zenodo.11528930)[92].

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

## Acknowledgements

This research was partially funded through a Doctoral Training Programme grant from NERC ENVISION (J.H.H. grant number: NE/L002604/1). N.A.J.G. is funded by a Royal Society fellowship (URF\R\201029). J.E.C. is supported by the Australian Research Council. J.P.W.R. was funded by a Leverhulme Trust Early Career Fellowship and Royal Society fellowship (URF\R1\231087).

## Author contributions

J.H.H., N.A.J.G., and D.M. conceived the study. I.R.C. and J.E.C. managed the database. J.H.H. analysed the data with support from A.C.S., D.R.B., B.M.T., N.M.D.S., I.R.C., J.P.W.R. The first draft of the manuscript was prepared by J.H.H. and all authors contributed to revisions. J.E.C., D.R.B., D.M., J.P.W.R., and N.A.J. provided editorial guidance and supervision.

## Competing interests

The authors declare no competing interests.
