## [Peer Review File · Nature Communications]

Global patterns and drivers of fish reproductive potential on coral reefsREVIEWER COMMENTS

Reviewer #1 (Remarks to the Author):

This paper seeks to quantify the effect of various anthropogenic and environmental factors on the future productivity (i.e. reproductive potential) of coral reef fish communities. This is an important topic, and is likely to be of interest to both applied conservationists/fisheries scientists and more basic-research population/community biologists. The approach taken in this MS is to estimate the total number of eggs per unit area in all reproductive female fish (of all species present) at a snapshot in time, based on data collected from observational surveys of 1,632 coral reef sites, located across the world. These estimated values of reproductive potential are then analysed to compare differences between marine protected / non-protected areas, and to explore the effects of various anthropogenic and environmental effects on reproductive potential. The results with respect to these effects are in agreement with those reported in Cinner et al (2016), from which this study draws its observational dataset. For example, the standardised effect sizes in Fig 2 in this MS show similar patterns to those seen in Box 1 in Cinner et al, although the effect sizes in this MS tend to be larger. One main difference in this MS is that instead of biomass (as in Cinner et al), the focus is on the reproductive potential of different coral reef sites. Another difference is that this study extrapolates its results to predict the impact of extending reef-protected areas to the 30% goal set by the UN Biodiversity Conference of the Parties in 2022.

I admire the diligence and thought that has gone into estimating the reproductive potential of different sites. The authors start from observations of length, number, and species of individual fish observed on coral reef surveys. They then estimate biomass, use the hyperallometric fecundity model of Barneche et al (2018) to estimate individual fecundity, then estimate the proportion of mature females of each species within the community, accounting for the effects of temperature differences between sites on maturation rate. This allows the estimation of reef-level reproductive potential, which is used as the response variable in a model assessing the impact of anthropogenic and environmental drivers.

Given the multi-step process through which reproductive potential is estimated, I think it is

worth giving a little more consideration/discussion to the underlying sources (and sinks) of variation that contribute to this metric. The first source of variation is in the underlying actual data - counts, lengths, and species identity of individual fish from observations on coral reefs. This captures real biological variation (plus measurement error), among individuals and among sites/communities. In the next stages, lengths are converted to predicted fecundity via an intermediate biomass conversion. This introduces new sources of variation in reproductive potential: differences in species-specific intercepts of the fecundity model mean that the reproductive potential of sites will differ if they have a different community structure. Given that the size-scaling coefficients are generally >1 , larger fish will contribute disproportionately to reproductive potential, meaning that differences in size-structure between sites will contribute to differences in reproductive potential between sites. I like how in lines 178-208 of the MS, the relative contributions of species composition and size structure differences (i.e. sources of variation) to differences in reproductive potential across sites are discussed. These are quite insightful, and I can see this being of interest to conservationists and fisheries scientists.

However, I am concerned about the lack of discussion around potential sinks of variation, i.e. unaccounted uncertainty that potentially make the results appear stronger than they are. One thing that isn't clear to me is how variation among individuals in fecundity is accounted for. Real fecundity data is extremely variable. In these calculations, are all fish of the same size and species expected to have exactly the same fecundity? If so, this will lead to anti-conservative results downstream, which should be explicitly addressed in the discussion. Another assumption here is that individual fecundity is entirely determined by size, i.e. there are assumed to be no effects of density-dependence, resource competition, etc (factors which are likely to differ between protected vs non-regulated reefs) on the form or parameterisation of the fecundity function. Since this could impact the predictions made, it is worth discussing the implications of this assumption.

On a related note, I am concerned about not accounting for the uncertainty in input parameters. Specifically, it isn't clear that the uncertainty in the parameters from Barneche et al 2018 - which are used to convert biomass estimates into fecundity estimates - are accounted for in the analyses. Are point estimates used for the calculation of fecundity?

Note that for Serranidae, the size-fecundity scaling coefficients reported in Table S3 of Barneche et al 2018 have considerable uncertainty: although more than half of the scaling coefficients for this family have values greater than or equal to the grand mean of 1.19, none are significantly different from 1, and half are based on sample sizes of fewer than 5 measured individuals. My worry is that the eye-catchingly huge effect sizes reported in this manuscript for Serranidae are based on statistically insignificant parameter estimates (at least when the null model is that reproduction scales proportionately to biomass, i.e. $\beta=1$). I think the manuscript would be improved if the uncertainty in the parameters used to estimate reproductive potential were propagated through the analyses and so properly accounted for. Not doing so will inflate type-I error, which would be (i) bad, and (ii) a shame given the efforts gone so far to account for downstream error via the nice Bayesian workflow employed in this work. For a paper such as this, which is potentially of interest to policy makers given the global dataset used, I would want extra reassurance that predictions are made on the basis of statistically sound model estimates.

As a final note, it would have been nice to be able to take a look at the code and data while reviewing this manuscript. With this paper based so heavily on simulation, it would have been helpful when reviewing to take a look at the parameter values and code used. I realise this is not a requirement, but I generally like to submit my code and data along with my manuscript, as I find it ultimately reassuring (if initially a bit nerve-wracking) to have someone else cast an eye over my code.

Minor comments:

Lines 296-298: This is a really clear and helpful description of the process used to estimate reproductive potential - excellent!

Lines 317-318: Please clarify if these are the species-specific intercepts and slopes from Barneche et al.

Line 357-358: "the sex ratio was 6 females to 1 male, 60% of those records would be

sampled randomly as females”. Am I missing something, or should this example be 6 females to 4 males, or 86% sampled as females?

Supplementary Figure 1: I think the word “raw” is better replaced by “estimated”. The measures of biomass are estimated based on many assumptions about length/weight relationships, sex ratios, temperature-dependent size at maturity etc. I understand that on the y-axis the fecundity terms are based on variable-standardised estimates, but to use the term “raw” for the biomass estimates feels a bit misleading, given that biomass was not measured.

Reviewer #2 (Remarks to the Author):

This paper has an interesting idea, that comparing egg production of marine fishes in and out of fished areas is a worthy and informative management indicator and should be used to justify the protection of spatial reserves. I think this argument needs to be carefully fleshed out. The link between egg production and productivity is notoriously uncertain. The idea that protected areas can supply and replenish populations outside of reserves, or elsewhere is also highly uncertain (line 28-29), as I am sure the authors are aware, but the short format of this introduction leaves no room for nuance or justification of the approach. In general, the intro leaves out a lot of the material needed to contextualize the importance of the study and justify the methods.

My second major comment is that the assumptions of this approach rely on general scaling relationships that are purported hold for all coral reef fishes. I don't find the conclusions (We demonstrate that if 30% protection were achieved, there could be an increase of between 134-225% in the reproductive potential of Serranidae across marine realms, and an increase of up to 25% in the reproductive potential of all fish families) to be convincing. This result has not been demonstrated, it has merely been suggested by a model, which is ignores the uncertainty in each relationship/assumption going in to this calculation. Larger fish have hyperallometric fecundity, but there's quite a bit of variability in that relationship among species, which the authors found in their conclusion that the exponent for reproductive potential was closer to 1 than expected (paragraph starting on line 178). This

demonstrates that there is a lot more complexity that this approach ignores.

In summary, the authors are making specific predictions based on visual surveys of reefs and then relying on high level macro-ecological relationships to infer family-level population metrics. It could be fine if pitched as a first step toward managing reproductive potential of target populations, but oversells the conclusions in the main text. For example, the authors extrapolate their results to suggest that protection of spatial reserves can improve biomass production (line 225), ignoring that the very strong assumption that fishing pressure on high-value species is not displaced outside of reserves - is unrealistic.

line 233 - I don't know what you mean by "traits that could be unaccounted for by including a phylogenetic covariance matrix"

line 241 - this would have been great material for the introduction

Line 312 - b should be exponentiated in this equation.

We would like to thank the reviewers for carefully assessing our paper and providing useful and constructive feedback. We have substantially improved the manuscript's analysis and text through revisions. Please see responses (**bold**) to specific points (**red**) below.

Reviewer #1 (Remarks to the Author):

This paper seeks to quantify the effect of various anthropogenic and environmental factors on the future productivity (i.e. reproductive potential) of coral reef fish communities. This is an important topic, and is likely to be of interest to both applied conservationists/fisheries scientists and more basic-research population/community biologists. The approach taken in this MS is to estimate the total number of eggs per unit area in all reproductive female fish (of all species present) at a snapshot in time, based on data collected from observational surveys of 1,632 coral reef sites, located across the world. These estimated values of reproductive potential are then analysed to compare differences between marine protected / non-protected areas, and to explore the effects of various anthropogenic and environmental effects on reproductive potential. The results with respect to these effects are in agreement with those reported in Cinner et al (2016), from which this study draws its observational dataset. For example, the standardised effect sizes in Fig 2 in this MS show similar patterns to those seen in Box 1 in Cinner et al, although the effect sizes in this MS tend to be larger. One main difference in this MS is that instead of biomass (as in Cinner et al), the focus is on the reproductive potential of different coral reef sites. Another difference is that this study extrapolates its results to predict the impact of extending reef-protected areas to the 30% goal set by the UN Biodiversity Conference of the Parties in 2022.

I admire the diligence and thought that has gone into estimating the reproductive potential of different sites. The authors start from observations of length, number, and species of individual fish observed on coral reef surveys. They then estimate biomass, use the hyperallometric fecundity model of Barneche et al (2018) to estimate individual fecundity, then estimate the proportion of mature females of each species within the community, accounting for the effects of temperature differences between sites on maturation rate. This allows the estimation of reef-level reproductive potential, which is used as the response variable in a model assessing the impact of anthropogenic and environmental drivers.

Thank you

Given the multi-step process through which reproductive potential is estimated, I think it is worth giving a little more consideration/discussion to the underlying sources (and sinks) of variation that contribute to this metric. The first source of variation is in the underlying actual data - counts, lengths, and species identity of individual fish from observations on coral reefs. This captures real biological variation (plus measurement error), among individuals and among sites/communities. In the next stages, lengths are converted to predicted fecundity via an intermediate biomass conversion. This introduces new sources of variation in reproductive potential: differences in species-

specific intercepts of the fecundity model mean that the reproductive potential of sites will differ if they have a different community structure. Given that the size-scaling coefficients are generally >1 , larger fish will contribute disproportionately to reproductive potential, meaning that differences in size-structure between sites will contribute to differences in reproductive potential between sites. I like how in lines 178-208 of the MS, the relative contributions of species composition and size structure differences (i.e. sources of variation) to differences in reproductive potential across sites are discussed. These are quite insightful, and I can see this being of interest to conservationists and fisheries scientists.

We thank the reviewer for these comments, and we are pleased they like our discussion in lines 220-222 regarding these sources of variation. This comment also prompted us to clarify how we account for variation in our models and discuss how these sources of variation have impacted our results. The first source of variation highlighted is that of fish surveys (counts, lengths, and species identity). We have added lines 459-465 in which we discuss how the hierarchical structure of our models captures inter-surveyor bias, as each reef cluster is surveyed by the same individual. Nevertheless, intra-surveyor measurement error is not easily captured, but it is reduced through training and experience, both of which the surveyors across our study have. This is now mentioned in the methods (lines 463-464). Conversion from length to biomass is a widely used and robust method for estimating fish biomass (MacNeil et al. 2015; Zamborain-Mason et al. 2023; Graham et al. 2020), and so we feel our existing discussion of the influence of size structure differences captures this element. Thank you also for stressing the importance of our findings in the light of the pressing imperative to protect 30% of the ocean before 2030.

Graham, Nicholas A. J., James P. W. Robinson, Sarah E. Smith, Rodney Govinden, Gilberte Gendron, and Shaun K. Wilson. 2020. 'Changing Role of Coral Reef Marine Reserves in a Warming Climate'. *Nature Communications* 2020 11:1 11 (1): 1–8.
<https://doi.org/10.1038/s41467-020-15863-z>.

MacNeil, M. Aaron, Nicholas A. J. Graham, Joshua E. Cinner, Shaun K. Wilson, Ivor D. Williams, Joseph Maina, Steven Newman, et al. 2015. 'Recovery Potential of the World's Coral Reef Fishes'. *Nature* 520 (7547): 341–44. <https://doi.org/10.1038/nature14358>.

Zamborain-Mason, Jessica, Joshua E. Cinner, M. Aaron MacNeil, Nicholas A.J. Graham, Andrew S. Hoey, Maria Beger, Andrew J. Brooks, et al. 2023. 'Sustainable Reference Points for Multispecies Coral Reef Fisheries'. *Nature Communications* 14 (1).
<https://doi.org/10.1038/s41467-023-41040-z>.

However, I am concerned about the lack of discussion around potential sinks of variation, i.e. unaccounted uncertainty that potentially make the results appear stronger than they are. One thing that isn't clear to me is how variation among individuals in fecundity is accounted for. Real fecundity data is extremely variable. In these

calculations, are all fish of the same size and species expected to have exactly the same fecundity? If so, this will lead to anti-conservative results downstream, which should be explicitly addressed in the discussion. Another assumption here is that individual fecundity is entirely determined by size, i.e. there are assumed to be no effects of density-dependence, resource competition, etc (factors which are likely to differ between protected vs non-regulated reefs) on the form or parameterisation of the fecundity function. Since this could impact the predictions made, it is worth discussing the implications of this assumption.

We thank the reviewer for these constructive comments. We adjusted our models to account for more sources of variation including intraspecies variability. In the new workflow, each fish species of a given size has a distribution of 1,000 possible fecundity values (lines 358-361). We did this by sampling 1,000 values from the posterior distribution of each extrapolated parameter, rather than taking the median value, to calculate each fecundity estimate. So, for example, a large *Plectropomus leopardus* weighing 5.8kg could have a fecundity ranging from $e^{20.5}$ eggs/female to $e^{10.1}$ eggs/female. Furthermore, we added a substantial discussion on the assumptions of our models (e.g. no density dependence) and the implications of these assumptions (lines 258-272).

On a related note, I am concerned about not accounting for the uncertainty in input parameters. Specifically, it isn't clear that the uncertainty in the parameters from Barneche et al 2018 - which are used to convert biomass estimates into fecundity estimates - are accounted for in the analyses. Are point estimates used for the calculation of fecundity? Note that for Serranidae, the size-fecundity scaling coefficients reported in Table S3 of Barneche et al 2018 have considerable uncertainty: although more than half of the scaling coefficients for this family have values greater than or equal to the grand mean of 1.19, none are significantly different from 1, and half are based on sample sizes of fewer than 5 measured individuals. My worry is that the eye-catching huge effect sizes reported in this manuscript for Serranidae are based on statistically insignificant parameter estimates (at least when the null model is that reproduction scales proportionately to biomass, i.e. $\beta=1$). I think the manuscript would be improved if the uncertainty in the parameters used to estimate reproductive potential were propagated through the analyses and so properly accounted for. Not doing so will inflate type-I error, which would be (i) bad, and (ii) a shame given the efforts gone so far to account for downstream error via the nice Bayesian workflow employed in this work. For a paper such as this, which is potentially of interest to policy makers given the global dataset used, I would want extra reassurance that predictions are made on the basis of statistically sound model estimates.

We appreciate the reviewer's suggestions to improve our model estimates by propagating uncertainty. We adopted the reviewer's suggestions by incorporating "measurement error" around the response variables in our models (lines 451-452; 476-481). In doing so, our uncertainty intervals incorporate more appropriate variation (e.g. see Figure 1.C.) and our estimates were amended due to shrinkage, so that estimates for sites with high measurement error values were corrected by

pooling information from more certain sites (McElreath 2018). Many effect sizes were also slightly reduced, but the important variables remained influential (e.g. full protection and market gravity).

McElreath, R., 2018. *Statistical rethinking: A Bayesian course with examples in R and Stan*. Chapman and Hall/CRC.

As a final note, it would have been nice to be able to take a look at the code and data while reviewing this manuscript. With this paper based so heavily on simulation, it would have been helpful when reviewing to take a look at the parameter values and code used. I realise this is not a requirement, but I generally like to submit my code and data along with my manuscript, as I find it ultimately reassuring (if initially a bit nerve-wracking) to have someone else cast an eye over my code.

We have made the code and data needed to reproduce the analyses available via GitHub at <https://github.com/Jeneen/ReproductivePotential>

Minor comments:

Lines 296-298: This is a really clear and helpful description of the process used to estimate reproductive potential - excellent!

Thank you

Lines 317-318: Please clarify if these are the species-specific intercepts and slopes from Barneche et al.

This has been clarified in lines 358-361.

Line 357-358: "the sex ratio was 6 females to 1 male, 60% of those records would be sampled randomly as females". Am I missing something, or should this example be 6 females to 4 males, or 86% sampled as females?

This was an error, thank you for the correction. See line 406.

Supplementary Figure 1: I think the word "raw" is better replaced by "estimated". The measures of biomass are estimated based on many assumptions about length/weight relationships, sex ratios, temperature-dependent size at maturity etc. I understand that on the y-axis the fecundity terms are based on variable-standardised estimates, but to use the term "raw" for the biomass estimates feels a bit misleading, given that biomass was not measured.

We agree that this is better terminology and have replaced "raw" with "estimated".

Reviewer #2 (Remarks to the Author):

This paper has an interesting idea, that comparing egg production of marine fishes in and out of fished areas is a worthy and informative management indicator and should be used to justify the protection of spatial reserves. I think this argument needs to be carefully fleshed out. The link between egg production and productivity is notoriously uncertain. The idea that protected areas can supply and replenish populations outside of reserves, or elsewhere is also highly uncertain (line 28-29), as I am sure the authors are aware, but the short format of this introduction leaves no room for nuance or justification of the approach. In general, the intro leaves out a lot of the material needed to contextualize the importance of the study and justify the methods.

We thank the reviewer for this feedback and have added more material and nuance to the introduction in order to contextualise the importance of the study and justify the approach taken (lines 23-42). We acknowledge that the assumption that protected areas can supply and replenish populations outside of reserves is indeed not always met and highlight this in the introduction (lines 46-47).

My second major comment is that the assumptions of this approach rely on general scaling relationships that are purported hold for all coral reef fishes. I don't find the conclusions (We demonstrate that if 30% protection were achieved, there could be an increase of between 134-225% in the reproductive potential of Serranidae across marine realms, and an increase of up to 25% in the reproductive potential of all fish families) to be convincing. This result has not been demonstrated, it has merely been suggested by a model, which ignores the uncertainty in each relationship/assumption going into this calculation. Larger fish have hyperallometric fecundity, but there's quite a bit of variability in that relationship among species, which the authors found in their conclusion that the exponent for reproductive potential was closer to 1 than expected (paragraph starting on line 178). This demonstrates that there is a lot more complexity that this approach ignores.

This was a useful criticism of the paper. We have incorporated more uncertainty into our models by including a "measurement error" term around our response variables (lines 451-452; 476-481). We have also toned down the language, making it more explicit that our results are based off model simulations (e.g. lines 202, 208). We acknowledge that other factors may be at play (line 249). For example, a time lag can be important between MPA establishment and an increase in fisheries yields (Barcelo et al. 2021), and the surroundings of an MPA (e.g. buffer zone of partial restrictions) can impact spillover effects (Ohayon et al. 2021).

Ohayon, S., Granot, I. and Belmaker, J., 2021. A meta-analysis reveals edge effects within marine protected areas. *Nature Ecology & Evolution*, 5(9), pp.1301-1308.

Barceló, C., White, J.W., Botsford, L.W. and Hastings, A., 2021. Projecting the timescale of initial increase in fishery yield after implementation of marine protected areas. *ICES Journal of Marine Science*, 78(5), pp.1860-1871.

In summary, the authors are making specific predictions based on visual surveys of reefs and then relying on high level macro-ecological relationships to infer family-level population metrics. It could be fine if pitched as a first step toward managing reproductive potential of target populations, but oversells the conclusions in the main text. For example, the authors extrapolate their results to suggest that protection of spatial reserves can improve biomass production (line 225), ignoring that the very strong assumption that fishing pressure on high-value species is not displaced outside of reserves - is unrealistic.

We appreciate this feedback and have added discussion around the implications of our assumptions and the effect this might have on the conclusions drawn from our models (lines 258-272). We provide suggestions for how future work could improve these limitations (lines 273-280).

line 233 - I don't know what you mean by "traits that could be unaccounted for by including a phylogenetic covariance matrix"

We have removed this from the discussion.

line 241 - this would have been great material for the introduction

We have moved this section to the introduction (lines 28-32).

Line 312 - b should be exponentiated in this equation.

This has been fixed.

REVIEWERS' COMMENTS

Reviewer #1 (Remarks to the Author):

I am happy that the authors have addressed my concerns, and appreciate that they have made the code available. My comments below are just suggestions.

Minor comments:

I have had a look at the code but couldn't run it - I suspect because the analysis was performed in R version 4.1.1, but my version is 4.2, and some of the packages are not available for 4.2. I couldn't open the generated data .rda files, nor use the .csv files which make use of Git LFS. This may be due to some conflict in my R libraries, but I would suggest that the authors check that someone other than the main coder can run the scripts and reproduce the analyses on a different machine, in case there are some missing dependencies. It would be useful if there were a main script, from which all the other scripts (31 of them!) were called, with annotation explaining what each one does. It would also be useful if the authors shared the output of their models in a file, so the output they generated can be checked without having to run everything again.

Lines 40-42: "While there are a multitude of uncertainties regarding the pathway from community reproductive potential to biomass production (e.g. larval survival, habitat availability, etc.), protection from fishing is likely to enhance both." I find this sentence confusing: protection from fishing enhances both what? Larval survival and habitat availability? The sentence reads as though protection from fishing enhances the "multitudes of uncertainty". Perhaps replace "a multitude of uncertainties regarding" with "many factors influencing"? And drop "both" unless the two examples come out of the parentheses? The reader expects the sentence to end with reference to the "multitudes" mentioned in the first part.

Reviewer #1 (Remarks on code availability):

I could not reproduce the analyses with the given code, possibly because of differences in the version of R I have installed (4.2) and that which the authors used (4.1.1). Additionally,

some of the code requires the use of an HPC, which I do not currently have access to.

It would be useful if there were a main script, from which all the other scripts (31 of them!) were called, with annotation explaining what each one does. It would also be useful if the authors shared the output of their models in a file, so the output they generated can be checked without having to run everything again.

Reviewer #2 (Remarks to the Author):

The authors have somewhat revised their analyses and text in response to my previous comments and those of Reviewer 1, who shared many of my concerns regarding the propagation of uncertainty in these figures. But the point made by Reviewer 1 that the "eye-catchingly huge effect sizes reported for Serranidae" are based on model assumptions derived from tenuous data still stands.

My concern arises from the way that these results stand to be interpreted by policy makers, because their robustness and generality are still overstated, in my opinion.

We would like to thank the reviewers for reassessing our paper after revisions and providing welcome feedback which we have now incorporated to improve our manuscript. Please see responses (**bold**) to specific points (**red**) below.

Reviewer #1 (Remarks to the Author):

I am happy that the authors have addressed my concerns, and appreciate that they have made the code available. My comments below are just suggestions.

Minor comments:

I have had a look at the code but couldn't run it - I suspect because the analysis was performed in R version 4.1.1, but my version is 4.2, and some of the packages are not available for 4.2. I couldn't open the generated data .rda files, nor use the .csv files which make use of Git LFS. This may be due to some conflict in my R libraries, but I would suggest that the authors check that someone other than the main coder can run the scripts and reproduce the analyses on a different machine, in case there are some missing dependencies. It would be useful if there were a main script, from which all the other scripts (31 of them!) were called, with annotation explaining what each one does. It would also be useful if the authors shared the output of their models in a file, so the output they generated can be checked without having to run everything again.

We really appreciate the reviewer's attempts to test our code. We have now:

- 1) Revised our code to use more up-to-date packages**
- 2) Updated our GitHub page to include more information on file structure**
- 3) Added an annotated main script which calls all the other scripts**
- 4) Run the analysis on two different computers to make sure that it runs smoothly**

Lines 40-42: "While there are a multitude of uncertainties regarding the pathway from community reproductive potential to biomass production (e.g. larval survival, habitat availability, etc.), protection from fishing is likely to enhance both." I find this sentence confusing: protection from fishing enhances both what? Larval survival and habitat availability? The sentence reads as though protection from fishing enhances the "multitudes of uncertainty". Perhaps replace "a multitude of uncertainties regarding" with "many factors influencing"? And drop "both" unless the two examples come out of the parentheses? The reader expects the sentence to end with reference to the "multitudes" mentioned in the first part.

We thank the reviewer for these comments and have now amended the text as per their suggestions.

Lines 39-43, Page 3: "While there are many factors influencing the pathway from community reproductive potential to biomass production (e.g. larval survival²⁵, habitat availability²⁶, etc.), protection from fishing is likely to enhance both reproductive potential and biomass production at independent points of the pathway²⁷."

Reviewer #1 (Remarks on code availability):

I could not reproduce the analyses with the given code, possibly because of differences in the version of R I have installed (4.2) and that which the authors used (4.1.1). Additionally, some of the code requires the use of an HPC, which I do not currently have access to.

It would be useful if there were a main script, from which all the other scripts (31 of them!) were called, with annotation explaining what each one does. It would also be useful if the authors shared the output of their models in a file, so the output they generated can be checked without having to run everything again.

We truly appreciate that the reviewer took the time to run through our code. In response to their comments, we have re-run the analysis on two separate computers to make sure the code runs smoothly and added a main script with annotations (as in response above).

Reviewer #2 (Remarks to the Author):

The authors have somewhat revised their analyses and text in response to my previous comments and those of Reviewer 1, who shared many of my concerns regarding the propagation of uncertainty in these figures. But the point made by Reviewer 1 that the "eye-catchingly huge effect sizes reported for Serranidae" are based on model assumptions derived from tenuous data still stands.

My concern arises from the way that these results stand to be interpreted by policy makers, because their robustness and generality are still overstated, in my opinion.

We thank the reviewer for their constructive criticism. We have addressed these points by:

- 1) Removing all mentions of model median estimates where they could not be expressed with their uncertainty (abstract) and expressing uncertainty where previously only median estimates were provided (discussion).**
- 2) Explicitly stated that our models were run with a set of assumptions and have very critical limitations that should be considered by conservation practitioners and policy makers (lines 221-227, 291-292).**

Lines 221-227, Page 11: "With all these results, it is critical to interpret median estimates within the context of their uncertainty and with the knowledge of our model assumptions and limitations. This is particularly important for conservation practitioners and policy makers wishing to assess the potential impact of protection, as our models make a series of assumptions (discussed below) and our methods were not able to account for a range of variables which are known to influence the success of protection (e.g. the size and age of protected areas, compliance with regulations, types of fishing restrictions, etc.^{4,43})."

Lines 291-292, Page 13: “These limitations to our approach mean that our results should be interpreted as an indication of the possible effects of conservation actions, rather than guaranteed outcomes.”

We highlight these assumptions throughout the discussion (e.g. lines 265-268, 277-285).

Lines 265-268, Page 12: “However, these scenarios were produced by randomly sampling sites to be protected and assuming fishing pressure was not displaced through leakage towards fishing grounds surrounding MPAs ⁴⁵. The location and selection of protected areas is critical to their success, both ecologically and socially ⁴⁶⁻⁴⁸.”

Line 277, Page 12: “However, our models do not account for population dynamics ⁵² and reproductive compensation mechanisms ⁵³...”

- 3) We suggest ways to improve future work and highlight the need for more comprehensive evidence directly quantifying the effects of fishing on reproductive potential (lines 293-300).**

Line 293-300, Page 13: “More comprehensive evidence quantifying the interacting effects of fishing and density-dependence on the reproductive traits of a diversity of species is needed to better improve estimations of reproductive potential. Future work could also account for differences in species’ life history/reproductive strategies that influence survival (e.g. reproductive care strategies) and lifetime reproductive output ⁵⁷⁻⁵⁹. Accounting for such variables will further increase the variation observed in reproductive potential between sites, but it could also increase our estimates of the impact of protection, particularly if protection has a direct impact, rather than an impact mediated through just biomass.”

- 4) We improved our analysis by ensuring our code is reproducible and uses a more up-to-date sampling function, thereby producing more reliable estimates, especially for Serranidae.**